# Investigation and Expression Analysis of R2R3-MYBs and Anthocyanin Biosynthesis-Related Genes during Seed Color Development of Common Bean (*Phaseolus vulgaris*)

**DOI:** 10.3390/plants11233386

**Published:** 2022-12-05

**Authors:** Musa Kavas, Mohamed Farah Abdulla, Karam Mostafa, Zafer Seçgin, Bayram Ali Yerlikaya, Çiğdem Otur, Gökhan Gökdemir, Aslıhan Kurt Kızıldoğan, Jameel Mohammed Al-Khayri, Shri Mohan Jain

**Affiliations:** 1Department of Agricultural Biotechnology, Faculty of Agriculture, Ondokuz Mayis University, Samsun 55270, Turkey; 2The Central Laboratory for Date Palm Research and Development, Agricultural Research Center (ARC), Giza 12619, Egypt; 3Department of Plant Biotechnology, College of Agriculture and Food Sciences, King Faisal University, Al-Ahsa 31982, Saudi Arabia; 4Department of Agricultural Sciences, University of Helsinki, PL-27, 00014 Helsinki, Finland

**Keywords:** MBW complex, anthocyanin synthesis, phenylpropanoid pathway, seed color, in silico

## Abstract

Anthocyanins are responsible for the coloration of common bean seeds, and their accumulation is positively correlated with the expression level of anthocyanin biosynthetic genes. The MBW (MYB-bHLH-WD40) complex is thought to regulate the expression of these genes, and MYB proteins, which are a key factor in activating anthocyanin pathway genes, have been identified in several plants. This study demonstrated gene structures, chromosomal placements, gene duplications of *R2R3-MYBs*, miRNAs associated with *R2R3-MYBs*, and the interaction of these genes with other flavonoid regulatory genes. qRT-PCR was used to investigate the role of specific *R2R3-MYBs* and flavonoid genes in common bean seed color development. As a result of a comprehensive analysis with the help of in silico tools, we identified 160 *R2R3-MYB* genes in the common bean genome. We divided these genes into 16 classes on the basis of their intron-exon and motif structures. Except for three, the rest of the common bean R2R3-MYB members were distributed to all chromosomes with different densities, primarily located on chromosomes 3 and 8. We identified a total of 44 duplicated gene pairs dispersed across 11 chromosomes and evolved under purifying selection (Ka/Ks  <  1), 19 of which were derived from a whole-genome duplication. Our research uncovered 25 putative repressor PvMYB proteins that contain the EAR motif. Additionally, fifty different cis-regulatory elements regulated by light, stress, and hormone were identified. Within the genome of the common bean, we discovered a total of 36 microRNAs that target a total of 72 R2R3-MYB transcripts. The effect of 16 *R2R3-MYB* genes and 16 phenylpropanoid pathway genes, selected on the basis of their interaction in the protein-protein interaction map, playing role in the regulation of seed coat color development was evaluated using qRT-PCR in 5 different tissues at different developmental stages. The results revealed that these specific genes have different expression levels during different developmental periods, with higher levels in the pod filling and early pod stages than in the rest of the developmental periods. Furthermore, it was shown that *PvTT8* (*bHLH*), *PvTT2* (*PvMYB42*), *PvMYB113*, *PvTTG1*, and *PvWD68* genes have effects on the regulation of seed coat color. The findings of this study, which is the first to use whole-genome analysis to identify and characterize the *R2R3-MYB* genes in common bean, may serve as a reference for future functional research in the legume.

## 1. Introduction

Common bean (*Phaseolus vulgaris* L.) is considered the second most important legume crop globally after soybean. In impoverished countries, the common bean seeds are among the most diverse protein sources and dietary minerals [1]. Similar to other grain legumes, the common bean seed contains various bioactive chemicals such as enzyme inhibitors, lectins, phytates, oligosaccharides, and phenolic compounds that play essential roles in metabolism in humans and animals who consume these foods regularly [2,3].

Flavonoids are believed to be essential for various physiological functions in plants throughout their lives. Flavonoids have been classified into several significant subgroups according to the nature of the C3 element, including chalcones, flavones, flavanols, flavan 3-ols, isoflavonoids, anthocyanins, and proanthocyanidins (PAs), anthocyanin, and flavonols are considered one of the main classes of flavonoids and are responsible for pigmentation in plants [4].

In Arabidopsis, most of the mutants created to reduce the amount of flavonoids were detected due to changes in seed pigmentation, and as a result of this mutation, transparent testa (tt) phenotypes were obtained [5]. Molecular evaluation of the tt loci has provided definitive information about the flavonoid biosynthetic pathway. Among the genes encoding flavonoid biosynthetic enzymes, two groups are distinguished: the early biosynthesis genes (*EBGs*) (*CHS*, *CHI*, and *F3H*) and late biosynthesis genes (*LBGs*) (*DFR*, *ANS*, *ANR*, and *LAR*) [6]. Interestingly, many studies have demonstrated a synergistic and robust correlation between the transcription of flavonoid biosynthetic genes with a ternary MBW complex. MBW complex includes *MYB* (myeloblastosis), *bHLH* (basic helix-loop-helix), and a WD-repeat (*WDR*) protein. In addition, these proteins can mutually interact, indicating that they bind target genes’ promoters to activate their transcription in planta [7].

MYB superfamily proteins are one of the most common and largest transcription factor families in eukaryotes. It is also worth noting that MYB superfamily proteins have a variety of roles in plants, ranging from regulation of secondary metabolite production and meristem formation to cell cycle control. Typical MYB domains in plants are composed of incomplete tandem repeats of 50–53 amino acid residues, which encode the helix-loop-helix structure [8]. The third helix is distinct as a recognition helix that directly contacts DNA and intercalates in the major groove [9,10]. MYB proteins are divided into four subfamilies, each of which is distinguished by the number of repeats located in the DNA-binding domain. These subfamilies include those with one repeat, also known as MYB-like proteins (R1MYB), two repeats (R2R3-MYB), three repeats (R1R2R3-MYB), and four repeats (R0R1R2R3-MYB). Plants, in contrast to the animal, encode MYB transcription factors with one to four repeats, whereas animals only encode MYB transcription factors with three consecutive repeats (R1, R2, and R3) [11,12,13,14].

Spatial and temporal patterning of anthocyanin in flowers, fruits, or vegetative tissues is controlled largely by R2R3-type *MYB* genes [15,16,17,18]. *R2R3-MYB* family members have been found and characterized in various plants. In the genomic sequence of *Arabidopsis thaliana*, there are 198 *MYB* family members, 126 of which encode two-repeat (R2R3) MYB proteins [10]. Over 110 R2R3–MYB proteins have been predicted in rice, while 244 R2R3–MYB proteins have been found in soybean (*Glycine max*) [11,19]. Many R2R3-type MYBs have been identified as transcriptional activators that activate gene expression by binding to cis-elements in targeted gene promoters. MYB suppressors were also recognized many years ago, but their abundance and importance have only recently been discovered [20]. While specific MYB repressor proteins in flavonoid biosynthesis have been found to bind basic helix-loop-helix factors and disrupt the MBW complex, it has been demonstrated that the lignin repressors MYBs interact with the *Cis*-elements of the promoter in target genes [21]. The function of the conserved repression motifs that characterize the MYB repressors, on the other hand, is still unknown [20].

In the present study, considering the various roles of MYB transcription factors, mainly their essential functions in flavonoid synthesis, the evolution and expression patterns of *R2R3*-type *MYB* genes in common beans were investigated using genome-wide analysis tools. In addition, expression analyzes of some flavonoid biosynthesis genes and genes encoding R2R3 type MYB transcription factors, which are in interaction in the protein-protein network, were performed using different tissues with different seed colors obtained from bean plants. This work aimed to find a gene associated with postharvest seed coat darkening in common beans and determine its role in promoting seed coat darkening.

## 2. Results

### 2.1. Identification, Classification, and Constructional Analysis of MYB Gene Family Members

As a result of the HHM search followed by keyword search in both the Plant Transcription Factor database and Phytozome v13, we initially identified 181 proteins carrying R2R3-MYB domains. However, 21 protein sequences were not considered true R2R3-MYB proteins because these domains were either missing or had another domain. Finally, 160 *PvMYB* genes encoding the R2R3-MYB proteins in the common bean genome were selected for use in subsequent analysis. Since a gene in *Arabidopsis thaliana* can have more than one ortholog gene in the common bean genome, the original gene names in Arabidopsis were not used, and the nomenclature PvMYB, starting from 1 and ending at 160, was used instead. Genomic and protein characteristics of these PvMYBs in *P. vulgaris* and analysis are given in Table 1 and Appendix A along with their chromosomal locations and gene IDs. It was observed that the *PvMYB* genes identified in the genome data of the common bean were significantly different in terms of gene and protein structures. There was a 9.439 bp difference among the longest (10.049 bp-*PvMYB34*) and shortest (610 bp-*PvMYB01*) members of *R2R3-MYB* genes. In the same manner, the number of amino acids (a.a) of PvMYBs varied between 185 a.a (*PvMYB01*) and 1626 a.a (*PvMYB91*), and their corresponding molecular weights (MWs) ranged from 20.28 (*PvMYB01*) to 176.96 (*PvMYB91*) kDa. The predicted isoelectric point (pI) values of PvMYBs in common bean changed from 4.96 (*PvMYB68*) to 9.87 (*PvMYB01*). The results of the instability index indicated that PvMYBs were found as unstable in test tube (instability index > 40) except for *PvMYB54* and the grand average of hydropathicity (GRAVY) suggested that all PvMYBs were hydrophilic proteins (GRAVY < 0) (Appendix A). Moreover, among PvMYBs in common bean, all of them were predicted to be located in the nucleus. Also, *PvMYB32* is expected to have chloroplast transfer peptide (0.2016) (Figure 1).

### 2.2. Phylogenetic Relationship, Conserved Motifs, and Gene Structures of the PvMYB Gene Family Members

It was discovered that *PvMYB* genes were evenly dispersed throughout all chromosomes (Figure 2). However, it was determined that 3 *PvMYB* genes, *PvMYB158*, *PvMYB159*, and *PvMYB160*, could not be distributed to the chromosomes and remained on the scaffolds. Among the 11 chromosomes in the bean genome, chromosomes 3 and 8 had the largest number of *PvMYB* genes, with 20 genes in each of these chromosomes. Furthermore, chromosome number 10 had the fewest *PvMYB* genes of any of the other chromosomes (Figure 2).

To elucidate the evolutionary mechanism among *PvMYB* genes in common beans, we constructed a phylogenetic tree with amino acid sequences of 160 putative PvMYBs. As shown in Figure 3, we grouped PvMYBs into 16 clades based on their sequence similarity. The sub-group A, L, and P were the smallest with a single *PvMYB* gene in each, and sub-group E was the largest among PvMYBs with 25 *PvMYB* genes (Figure 3). The analysis showing the gene structure confirms the phylogenetic tree because the genes in the same subgroup have similar intron-exon numbers. These analyses also revealed that *PvMYB1, PvMYB18, PvMYB52, PvMYB119, PvMYB87*, and *PvMYB125* genes are intronless. In addition, it is noteworthy that introns are located in the 5′-UTR of *PvMYB11*, *PvMYB137*, *PvMYB138*, and in the 3′-UTR region of *PvMYB2* and *PvMYB126*. The highest number of introns are found in the *PvMYB23* and *PvMYB40* genes, with 13 introns each. One hundred five of the analyzed R2R3-type *MYB* genes in common bean contain two introns, which constitute the largest group. We concluded that R2R3-type MYB members in the same branch have similar motifs, which supports the phylogenetic tree, by the analysis we made with the MEME suite. As a result of this analysis, it was observed that there are at least three and at most eight motifs in the members of the MYB protein family (Figure 4).

### 2.3. Analysis of Evolutionary Divergence in PvMYB Transcription Factors

To better understand the evolutionary divergence of common bean R2R3-MYB TFs, we analyzed the synteny relationship of all *PvMYB* genes against model species, Arabidopsis, and rice genomes using MCScanX in TBtools and visualized using SynVisio (Figure 5, Appendix A). In total, out of 160 *PvMYBs*, we detected 141 genes with an orthologous relationship to *A. thaliana*, whereas 47 *PvMYB* genes were found orthologous with O. sativa. In addition, 24 *PvMYB* genes were found to share ortholog to both *A. thaliana* and *O. sativa*, showing these genes’ common ancestry.

Gene duplication analyses were performed on PvMYB TFs within common bean chromosomes. We found 44 gene duplication events distributed on 11 chromosomes (Figure 6). In detail, 19 gene pairs with whole-genome duplication, 17 pairs with dispersal duplication, 4 *PvMYB* gene pairs with tandem duplication, three pairs with transposed duplication, and only one pair with proximal duplication were identified according to Plant Duplication Gene Database (PlantDGD). Interestingly, 10 *PvMYB* genes, including *PvMYB11, PvMYB83, PvMYB86, PvMYB101, PvMYB120, PvMYB123, PvMYB124, PvMYB144, PvMYB146*, and *PvMYB154*, were found to be present in seven syntenic duplicated gene pairs. This finding highlights the importance of these genes in the expansion of the R2R3-MYB transcription factors in common beans.

The Ka/Ks values presented in Appendix A showed a wide range of values indicating the presence of selective pressure during gene duplication among *PvMYB* genes. The highest was 0.625, shown between *PvMYB36* and *PvMYB37*, and the lowest value was recorded as 0.0987 between *PvMYB18* and *PvMYB126*. All PvMYBs had Ka/Ks ratios <1, indicating that they may have undergone purifying or negative selection. We further used the Ks value to calculate the divergence time of the 44 duplicates in Million Years Ago (Mya). As indicated in Appendix A, the divergence times of *PvMYB* genes were widely diverse, with the earliest duplication event predicted to occur at 34.99 Mya and the most recent duplication occurring at 0.20 Mya. In contrast, the average value was recorded as 11.47 Mya.

According to previous studies, those proteins containing the EAR or TLLLRF motifs can act as transcriptional repressors by interacting with co-repressors and altering the structure of chromatin through histone modifications, thereby repressing gene transcription [22]. On the basis of our analysis, we were able to identify 25 PvMYB proteins, assuming that these proteins function as repressors. Appealingly, all these proteins contained the EAR motif, and none of them included the TLLLRF motif. Moreover, the EAR motifs were defined only by the consensus sequence patterns of LxLxL, not by DLNxxP patterns. All the predicted 25 R2R3-MYB proteins had a single EAR motif in their amino acid sequence except for PvMYB54 protein, which had a pair of EAR motifs on both terminals. Additionally, the position of these motifs in the amino acid sequence is indicated as N-terminal, mid-terminal, or C-terminal region (Appendix A). There are 17 PvMYB proteins that have the EAR motif on the C-terminus and others located at N-terminals. The alignment of the truncated amino acid sequences illustrated the core motif EAR (LxLxL pattern) for our putative repressor proteins (Appendix A).

### 2.4. Prediction of Cis-Elements and miRNAs Targeting PvMYB Genes

In the current study, we identified regulatory elements in the promoter region of selected genes encoding R2R3-MYBs and located in flavonoid and phenylpropanoid pathways to figure out their transcriptional regulation leading to common bean seed color. Fifty different cis-regulatory elements regulated by light, stress, and hormone were identified in the putative promoter regions of these target genes (Appendix A, Appendix A). Among them, 22 hormone-responsive cis-acting elements were predicted, and the most abundant elements are ERE (Ethylene-responsive element) and ABRE (ABA-responsive elements). Secondly, 19 cis-acting regulatory elements were expected to be associated with the light-response, in which BOX-4 and G-box were found highly and significantly among target genes’ putative promoter sequences. Specifically, ARE (antioxidant-responsive element), MYB, and MYC, among 19-stress regulatory elements, were mainly predicted, but MYB and MYC acting-regulatory elements were significantly found in the putative promoter region of all target genes.

Additionally, we identified a total of 36 miRNAs targeting 72 R2R3-MYB transcripts in the common bean genome. As shown in Appendix A, 23 PvMYB transcripts had at least two miRNA target sites. *PvMYB160* had the highest miRNA targeting sites in which five miRNAs showed complementary to five different sites on the transcript. Next were *PvMYB79* and *PvMYB80*, each being targeted with four different miRNAs. Eight PvMYB transcripts (*PvMYB68/87/93/117/123/133/139* and *PvMYB156*) had three miRNA target sites in their sequences. Twelve PvMYB transcripts were predicted to consist of a double, whereas 49 PvMYBs had single miRNA target sites. Among the 36 miRNAs targeting the PvMYB mRNA in common bean, Pvu-miRN2588 was shown to have the highest number of targeted sites, potentially cleaving to 13 different PvMYB transcripts. Pvu-miR395, Pvu-miR319, and Pvu-miR159 were next, with 11, 10, and 7 targeting regions in PvMYBs, respectively. Pvu-miR396 and Pvu-miRN2600 were complementary to six sites present in sequences of six different PvMYBs.

In the current study, 10 out of 32 target transcripts, their activities analyzed by qRT-PCR, were found as targeted by 16 different miRNAs based on the complementarity of their transcript sequences (Appendix A). Among them, the highest number of miRNA sites were identified in *PvMYC1* and *PvMYB93* (three different in both). In addition, the transcripts of *PvF3’5’H, PvF3G*, and *PvMYB113* were found to be potentially targeted by two other miRNAs each. Appendix A indicates that only the Pvu-miR397 has two different targets in the transcripts of the core gene set.

### 2.5. Analysis of the Core Gene Set in Common Bean (P. vulgaris) Seed Color

The protein-protein interaction (PPI) network was created using the genes involved in flavonoid/phenylpropanoid pathways and PvMYBs with a potential role in common bean seed color (Appendix A). The protein-protein interaction analysis indicated a strong protein network of clear biological connectivity (PPI enrichment *p*-value: < 1 × 10^−16^). The interaction network consisted of 79 edges and 36 nodes. Additionally, the interactions between nodes referenced high confidence scores (ranging from 0.723 to 0.979). The PPI network was clustered into three main clusters by specified MCL inflation parameters. XP_007148576.1, XP_007153909.1, XP_007162416.1, XP_007137926.1 and XP_007141539.1 was found as predicted functional partners to construct this PPI network.

We compared the expression levels of 32 genes, selected from the PPI network, between different common bean cultivars (Seminis (Control group), Altın, Perola, Atlantis, Dark red kidney, and Çarşamba) by qRT-PCR (Figure 7 and Figure 8). The results showed that the differences in the flesh color phenotype of common bean grains among the six cultivars are significantly induced by MYB regulation and flavonoid biosynthesis, especially in 3DT and 4DT. *PvMYB146* was highly upregulated in Çarşamba (156.5-fold), Atlantis (113-fold), Dark red kidney (58.5-fold), and Perola (57.3-fold) cultivars at the early pod stage of development. This gene expression level order correlates well with the seed coats’ pigment content (Its expression increased in the order of pigmentation level). In these cultivars, the *PvMYB146* expression sharply decreased at the pod stage, even stopping in the Perola and Dark Red Kidney cultivars. In general, the buds of the same four cultivars expressed this gene in a range of 1.7 to 5 times more than the Seminis buds (Figure 7).

The *PvMYB146* showed 1.5 and 1.6 times more expression in the Altın cultivar at early pod leaves and pod fill stages, respectively. Likewise, *PvMYB131* exerted a similar but lower expression pattern in these cultivars except for the Altın cultivar. In the Altın cultivar, the *PvMYB131* gene expressed 9.4 times higher in the pod fill stage.

Among the MYB encoding genes, *PvMYB149*, *PvMYB142*, *PvMYB141*, *PvMYB93*, and *PvMYB92*, the highest level of expression corresponded to the early pod stages of Atlantis cultivar ranging from 109.9 to 279.2-fold increase. The Çarşamba cultivars at the same developmental stage followed this high expression level (79.1- to 278.2-fold higher expression). At the pod fill stage, the Altın cultivars expressed these five genes at high levels (8.7- to 37.8-fold). Although there were higher expression levels in the darkest cultivars, Atlantis and Çarşamba, no direct correlation was recorded with the level of pigmentation for these related genes. Interestingly, *PvMYB94* was the most upregulated gene among the tested five genes in the Altın cultivar corresponding to 1199.1-fold higher expression in the pod fill stage. Its expression also slightly increased in the developmental stage in the Dark red kidney, Atlantis, and Çarşamba. In general, MYB-related genes showed increased expression levels in this cultivar later than the remaining ones (Figure 7).

The *PvMYB128, PvMYB56*, and *PvMYB50* showed an increased expression pattern by the darkness of the flesh color of the grains in the early pod stage in all cultivars. However, Atlantis exerted a higher gene expression level in the following stage. The expressions of the *PvMYB50, PvMYB19*, and *PvTT8-1* provided quite similar expression patterns to others (*PvMYB128, PvMYB56*, and *PvMYB50*). With only one exception that their higher expression levels slightly decreased in the Çarşamba cultivar, showing the highest expression in the Atlantis.

Among the flavonoid-related genes studied, the *Pv4Cl* was the most upregulated gene, with a 152.9-fold increase in Atlantis and 208.1-fold in Çarşamba cultivars at the early pod stage. Its expression level was also higher in the leaves of the early pod stage (70.6 times more expression). A higher level of expression, ranging from 75.04- to 93.2-fold, in the early pod stage for the Atlantis and Çarşamba cultivars was recorded by *PvF3’M, PvF3’5’H, Pv2OGFeDO*, and *PvGT1* (Figure 8). These genes are responsible for the biosynthesis of flavonoid 3′-monooxygenase (E1.14.13.21), flavonoid 3′,5′-hydroxylase (CYP75A), 2-OXOGLUTARATE (2OG), and FE(II)-DEPENDENT OXYGENASE SUPERFAMILY PROTEIN-RELATED and Anthocyanidin 3-O-glucosyltransferase/Uridine diphosphoglucose-anthocyanidin 3-O-glucosyltransferase enzymes, respectively. The other genes involved in flavonoid biosynthesis such as *PvFLS1, PvWDR68, PvPAL*, and *PvCLL* encoding for Naringenin-chalcone synthase/Flavanone synthase WD repeat-containing protein 68 (WDR68, HAN11), Phenylalanine ammonia-lyase, and 4-COUMARATE--COA LIGASE-LIKE 6 enzymes, respectively, were highly upregulated in the Dark red kidney cultivar (96.8-, 25.3-, 35.3- and 50.4-fold respectively), then in the Atlantis and Çarşamba cultivars with the lower expression levels in the early pod stage. However, the flavonoid-related genes in this study exhibited remarkable expression levels, especially in the leaves of the early pod stage. For instance, the *PvGT1* expression level was 25-fold higher in the leaves of the early pod stage in the Çarşamba cultivar. The Atlantis cultivar expressed at its highest level (26.8-fold) in the early pod stage of the *PvANR* gene, which is responsible for the biosynthesis of anthocyanidin reductase (ANR) in this category (Figure 8).

## 3. Discussion

Genome-wide studies give us an opportunity to understand and characterize the possible functions of genes or gene families that share specific protein domains and motif structures in an organism. R2R3-MYB is a member of the MYB transcription factor family, which is distinguished by its highly conserved R2R3-MYB domains. In most plants, members of this family are involved in anthocyanins and PA biosynthesis. Besides, basic helix-loop-helix (bHLH) and TRANSPARENT TESTA GLABRA1 (TTG1) (WD40 proteins) form ternary complexes (‘MBW’). The discovery of the type, and spatiotemporal activity of these MBW complexes, and identifying direct targets are critical steps toward increasing knowledge of the transcriptional processes that drive anthocyanins and PA biosynthesis. MYB transcription factors have been functionally investigated in as many as 74 plant species, and most studies focused on the R2R3-MYBs, which are one of the most prominent transcription factor gene families. These include plant species such as *A. thaliana, Z. mays, P. trichocarpa, S. tuberosum*, and *Gossypium spp*. [14,23,24].

### 3.1. R2R3-MYB Proteins Associated with Seed Coat Color in P. vulgaris

As natural pigments, anthocyanins are responsible for blue, purple, red, and orange colors in many flowers, fruits, seeds, and vegetables. Accordingly, the primary purpose of the present study was to unriddle the obscurity bearing between *R2R3-MYB* genes and their direct targeted flavonoid biosynthesis-related genes with seed coat color in common beans. This paper identified 160 members of the *R2R3-MYB* genes in the common bean plant which were systemically named. There is considerable variation in the number of genes encoding R2R3-type MYB proteins in plants. In this context, 126 genes have been identified in *Arabidopsis thaliana* according to previous studies [10]. Likewise, more than 110 *R2R3-MYB* candidate genes were predicted in rice, while 244 *R2R3-MYB* genes were found in soybean (*Glycine max*), 118 in grape, and 192 in poplar [11,19,25,26]. The main variation in gene numbers can be attributed to the size of the investigated plant’s genome. Additionally, many *PvMYBs* are putative orthologs of a single *Arabidopsis MYB* gene, suggesting that MYB genes in common beans experienced duplications following the divergence of common bean and *Arabidopsis*. Likewise, the first Whole Genome Duplication (WGD) event in the genome of soybean and common bean occurred approximately 56 million years ago, and these two organisms diverged from each other about 19 million years ago. After this separation, the second WGD event happened in the soybean about 10 million ago. Due to these duplications, which occur twice, soybean has 244 R2R3 MYBs in their genome, while beans have 160 [27,28].

The physicochemical properties of R2R3-MYB proteins were investigated in order to gain a better understanding of their nature. There is a similarity between all the members of R2R3-MYB in the GRAVY scores, which indicates that all R2R3-MYB proteins had negative GRAVY values, which means they are hydrophilic. When examined in terms of physicochemical properties, it is thought that a protein with an average hydropathy score greater than zero may be an integral membrane protein, while one with a negative value may be in a globular structure [29]. We realized that a negative GRAVY score is the most dominant for R2R3-MYB proteins in common, showing that MYBs are soluble proteins, which is a characteristic requirement for transcription factors. Proteins are sorted and transported into appropriate subcellular sites in living cells to fulfill their biological functions [30]. Knowing a protein’s subcellular localization site can be very useful in determining its cellular function. Different localization predictor software were used to predict the subcellular localization of R2R3-MYB proteins. The results revealed that 99.38% of R2R3-MYB proteins were located in the nucleus; our result has been supported by a similar study conducted in rice by Katiyar et al. [19].

The organization of *MYB* genes’ exons and introns was studied to comprehend their structural components better. We determined that six genes were intronless (Figure 4). As mentioned in numerous pieces of literature, intronless genes are a distinctive trait of prokaryotes, and most eukaryotic genes are interrupted by one or more non-coding regions termed introns. However, studies on intronless genes in eukaryotes have been published in the last few decades [31,32]. Intronless genes in eukaryotes provide fascinating datasets for comparative genomics and evolutionary investigations. Illustrating these genes can be an advantage in understanding the evolutionary patterns of related genes and genomes [33]. Katiyar et al. [19] also reported that (10.96%) *OsMYB* and (4.56%) *AtMYB* genes were intronless. In addition, most *PvMYB* genes have the highly conserved splicing pattern found in other plant species, and they conform to the rule reported earlier, that there would be two introns and three exons in a gene. Genes from the same subfamily have nearly identical intron patterns, with the intron location almost fully conserved across most subfamilies. Previous research suggested that an intron-rich gene would lose many introns simultaneously due to retro-transposition, resulting in intron-free ancestral genes [24].

### 3.2. Prediction of Putative Suppressors among PvMYB Genes

Detailed analysis of MYB suppressor proteins has only recently been performed. MYB repressors frequently contain repressive motifs, such as the EAR motif and the TLLLFR motif, which are involved in the complex regulatory mechanisms that repress genes involved in anthocyanin production. The study of transcriptional repression necessitates the prediction and functional annotation of EAR motif-containing proteins. Thus, we sought additional evidence to predict and examine the candidate repressors’ *R2R3-MYB* genes. We identified 25 R2R3-MYB proteins with the assumption that these proteins function as a repressor. Similar results have also been reported by [20,34].

### 3.3. miRNAs Targeting Expression of PvMYB Genes

microRNA (miRNA) is a non-coding, small single-stranded RNA endogenous to an organism and possesses an extensive range of biological functions. Their involvement in plant growth and developmental regulation, hormone, and immune response, floral organ regulation, and biotic and abiotic stress responses are widely studied [35,36]. miRNAs are also investigated in their role in post-translation gene regulation of flavonoid biosynthesis and other secondary metabolites [37]. However, the roles of miRNAs in targeting R2R3-MYB responsible for common bean seed coat color are still lacking. Our study found that uracil (U) was the starting nucleotide in 72% of the predicted miRNAs stated by Zhang et al. [38], that the positioning of U as the first nucleotide greatly depends on miRNA-mediated regulation. He et al. [39] and Robert-Seilaniantz et al. [40] reported that miR393 cleavage would confer down-regulation in the expression of flavonoid key genes. We predicted that Pvu-miR395 had cleavage and translation inhibition in 11 *PvMYB* genes; miR393 had three target sites, while miR394 had two target sites. This might indicate repression in the expression of these 16 *PvMYB* genes in the common bean. In a comprehensive study conducted by Gupta et al. [41] on the regulatory mechanisms of miR319, miR159, miR156, and miR396, which are also targeting our genes of interest, were reported to play an important role in the regulation of flavonoid biosynthesis accumulation probably by destabilization of an MYB-bHLH-WD40 transcriptional activation complex. miR156 was also dubbed a master regulator of developmental transition in flowering plants; its regulations were recorded highest during early embryo and progressively reduced with plant age [42]. miR172, which targeted *PvMYB151*, was shown to target genes involved in phenylpropanoid biosynthesis along with other secondary metabolite biosynthesis pathways [37,43]. miRNA390 was functionally studied along with miR156 and was found to be associated with the accumulation of trans-acting small interfering RNAs (tasiRNA), which target plant developmental genes [44]. miR390 targets *PvMYB34*, *PvMYB39*, and *PvMYB156*, indicating the importance of these genes during fruit ripening and plant development.

In the frame of 32 selected genes for expression analysis, which are involved in flavonoid biosynthesis, only ten genes (*PvMYC1, PvPAL, PvFLS2, PvMYB93, PvANR, PvTT8-1, PvF3G, PvMYB113, PvF3’5’H*, and *PvMYB56*) were found to be targeted with miRNAs. Sixteen miRNAs were found targeting 18 transcripts. In this context, *PvMYC1* was targeted by miR156 and miR157 in almost the same region of the transcript [39]. *PvMYB93*, highly upregulated in common bean, was targeted with three different miRNAs (Pvu-miR395, Pvu-miR319, Pvu-miR397), and according to our information, none of them have so far been reported to be associated with flavonoid biosynthesis.

Gene duplication in an organism is created by either whole-genome duplication (WGD) or single-genome duplication, which is in turn classified further into tandem, transposed dispersal, or proximal duplication [45]. These naturally occurring duplications are major factors in gene families’ generational maintenance and evolution [46]. Our study found a total of 44 gene duplication events in R2R3-MYB Tfs. Most of them (44.2%) demonstrated whole-genome duplication events, 38.6% were predicted to evolve by dispersal duplication, and only four gene pairs had tandem duplicates. This is in line with recent genome-wide investigations carried out on R2R3-MYB Tfs [10,47]. However, wheat R2R3-MYB Tfs had more WGD frequencies than common beans [48]. Low frequencies of tandem duplication events were also reported in *A. thaliana* by Cannon, Mitra [46]. Prior studies discussed thoroughly that *R2R3-MYB* genes regulate different floral pigmentation pattern elements and the role of gene duplication on those patterns [49,50,51,52]. We also performed a three-way collinearity analysis between *P. vulgaris* and *A. thaliana*, and *O. sativa*. We showed that the number of orthologs between *R2R3-MYB* genes in *P. vulgaris* and *A. thaliana* had the highest proportion compared to the number of orthologous genes between *P. vulgaris* and *O. sativa* (Figure 5). This result suggests that *P. vulgaris* is more related to *A. thaliana* than *O. sativa*. This might result from segmental duplication, which might accelerate the expansion of the gene family in the common bean.

### 3.4. Expression of Candidate PvMYB Genes and Anthocyanin Regulatory Genes Using RT-qPCR

R2R3-MYB transcription factors have been linked to regulating anthocyanin accumulation in plants, and some are implicated in upregulating genes present in the anthocyanin biosynthesis pathway, while other R2R3-MYB TFs are shown to repress their expression [53,54,55,56,57]. A further study proposed that three regulatory genes, *PvMYB1, PvMYB2*, and *PvTT8-1*, were dramatically upregulated in purple pod (kidney bean) skin compared to those found in other sources [58]. Our study showed that selected genes have different expression levels in different developmental periods, with higher levels in pod filling and early pod stages. For example, *PvMYB92, PvMYB93, PvMYB141, PvMYB142*, and *PvMYB149* genes showed a significant increase in expression in the early pod period in the Atlantis genotype. A study conducted on tomato showed the orthologs of *SlAN2*, and *SlANT1* genes, which encode MYB proteins, play a role in the regulation of anthocyanin synthesis [59]. As a result of our expression analysis, we determined that the expression of genes related to both MYB and anthocyanin synthesis in leaf tissue is relatively low compared to other tissues. A similar report was made by researchers investigating the effect of *MYB* genes on the color formation of sweet cherry fruit. This study determined that the *PavMYB10* gene, which is the ortholog of *PvMYB50*, controls color formation in fruit, but its expression level is almost non-existent in flowers and leaves [60]. Recent studies show that MYB-bHLH-WD40 (MBW) complex regulates anthocyanin biosynthesis. In this context, the expressions of *PvTT8 (bHLH), PvTT2 (PvMYB42), PvMYB113, PvTTG1* (WD40 repeat), and *PvWD68* (WD40 repeat) genes were also compared depending on the seed coat color changes. We observed a similar expression pattern of these genes in the same tissues of different genotypes.

Interestingly, apart from other *PvMYB* genes, the activity of the MBW complex gene members was observed in leaf tissue, particularly those aggregated at the formation of the early pod stage. These results are consistent with previous studies showing that the MBW complex controls anthocyanin biosynthesis in vegetative tissues of *A. thaliana* [61,62]. One of the critical enzymes in the phenylpropanoid pathway is phenylalanine ammonia-lyase (PAL). Although the transcript levels of the *PAL* gene expressed in the fruits of some tomato genotypes were reported to be high even at the ripening stage, it was determined that the PAL level decreased strongly after the fruit of a different genotype matured [63]. In our research, we came up with a similar conclusion. Because the expression of this gene is generally high in the early pod period, its activity in other tissues, including pod filling, was lower than at the early pod level.

## 4. Materials and Methods

### 4.1. Plant Material

The common bean varieties used in this study were obtained from the Middle Black Sea Transitional Zone Agricultural Research Institute. Six common bean cultivars were used according to the color of the seed coat: Altın, Çarşamba, Perola, Dark red kidney, Seminis, and Atlantis. Seminis is white, while Çarşamba seeds are the darkest ones. The color of seed coats in these cultivars darkens in the order shown in Figure 9.

Bean seeds were surface sterilized with a 5% Sodium Hypochlorite solution for 5 min, followed by a five-times rinse with sterile distilled water. Seeds were then sown in clean plastic pots (5.2lt) filled with soil-free vermiculite (approximately 5 kg for each pot) to ensure uniformity and the absence of substrate for microbial growth. In order to ensure a homogeneous germination in all pots, three seeds were planted in each pot and after homogeneity was achieved in the hypocotyl stage, one plant was left in each pot. Pots were irrigated once a week with a Hoagland’s solution. The seeds were grown in a controlled growth chamber set at 24 °C with a photoperiod of 16 h light and 8 h dark and humidity of 60–65%. Samples were collected at three different stages, including bud (1DT, 40 days after germination) and accompanied leaves (1Y), early pod (3DT, 55 days after germination) and accompanied leaves (3Y), and pod fill (4DT, 65 days after germination), were collected from all cultivars (Appendix A).

### 4.2. Expression Analysis Using Quantitative RT-PCR

Following the kit protocol, the total RNA isolation of all plant samples was performed using the QIAGEN RNeasy Plant Mini Kit. The total RNA concentration obtained at the end of the process was measured with the NanoDrop™ 2000c spectrophotometer device. For the expression analysis of target genes in genotypes having different seed colors, Seminis was selected as a control because of its colorless seed coat color. A total of thirty-two genes were chosen for qRT-PCR reactions, 16 of which belonged to the R2R3-MYB family, and the rest were targeted by R2R3-type MYBs in common beans based on previous literature data and protein-protein interaction. The qRT-PCR reaction was performed and calculated according to the 2^−ΔΔCt^ method as previously described [64,65]. *Pvactin11* (XM_007139153.1) was used as the internal housekeeping gene. The Primers used for this study were designed using NCBI primer Blast and are listed in Appendix A. Changes in relative expression levels of the target genes were checked for statistical significance in accordance with the one-way ANOVA. The means and standard deviation of the replications were compared by the least significant difference (LSD) test at the *p* ≤ 0.001.

### 4.3. Determination of MYB Members in the Common Bean Genome

R2R3-type *PvMYB* genes used in this study were identified according to previously described methods by Kavas et al., 2016 [66]. We first extracted protein sequences of R2-R3 MYB using Hidden Markov Model (HMM) downloaded from the Pfam database. This was then used as a query to search against the common bean genome using HMMER3.0. Putatively annotated MYB transcription factor genes in *P. vulgaris* from the plant TF database (http://planttfdb.cbi.pku.edu.cn, accessed on 6 November 2022) [67] were also used to check for protein sequences that might be excluded from the previous method. Then, we confirmed and carefully filtered each downloaded MYB protein sequence based on its motif structure. We then cross-checked the extracted *MYB* genes against Pfam and SMART databases to confirm the presence of the R2R3-MYB domain in all genes retrieved from the common bean genome v2.1. Protein sequences with two MYB domain repeats were accepted as the R2R3-MYB family as previously described by Stracke et al. [10]. The Phytozome v13 database was employed to further check the presence of all the *PvMYB* genes with their correct respective id names and to remove the duplicate sequences. The final sequences were numbered according to their position on the chromosome, starting from the first chromosome, numbers of which are shown in Appendix A, will be referred to as *PvMYBs* from now on.

The MEME suite tool (http://meme-suite.org/tools/meme, accessed on 6 November 2022) was used to evaluate the motif structure of PvMYB protein sequences. The number of repeats of each motif was set to 15 for this analysis [68]. Motifs are numbered based on their location in the protein sequence. The conserved domain was predicted by NCBI Conserved Domain Database (CDD) batch-search [69]. We analyzed the Exon-Intron structure on *PvMYB* genes using the GFF3 file downloaded from the Phytozome v13 database. Finally, we used Gene Structure View in TBtools software to combine and visualize motif organization, conserved domain, and exon-intron structure files [70].

The number of amino acids, length of the open reading frame (ORF), the isoelectric point of protein (pI), and molecular weight (MW) of all PvMYB proteins were analyzed using the ProtParam tool on the ExPASy web server (http://www.expasy.ch/tools/pi_tool.html, accessed on 6 November 2022) and Phytozome v13 web tool.

### 4.4. Chromosomal Position, GO Annotation, and Subcellular Localization in Common Bean

We predicted the subcellular localization of PvMYB proteins on a common bean using an online Target P-2.0 server [71]. The chromosomal position of *PvMYB* genes was analyzed using a generic feature format version 3 (GFF3) retrieved from the Phytozome v13 database. Then, TBTools software was used to map *PvMYB* genes onto their respective location on chromosomes. For GO annotation and in silico analysis of the *PvMYB* genes, we first launched a BLASTP search on NCBI and Nr databases with an e-value cut-off set at 1 × 10^−10^ with PvMYB amino acid sequences as a query. Then, we employed the Blast2GO program for the GO analysis with default parameters [72].

### 4.5. Identification of the Cis-Acting Elements on the Putative Promoter Region of Analyzed Genes

Two different gene groups were used for this analysis. In the study, a 2000 bp DNA sequence from the upstream region of the translation start site of the *PvMYB* gene was used. Required nucleotide sequences were retrieved from the Phytozome v13 database in Fasta format. Online software PlantCARE was used to analyze the functional putative promoter elements [73]. The second analysis using the same methodology was performed with some flavonoid synthesis-related genes, which were found to be associated with *R2R3 type MYB* genes from the protein-protein interaction analysis.

### 4.6. Prediction of miRNA Targeting PvMYB Genes

To analyze the miRNAs present in *Phaseolus vulgaris* targeting MYB transcripts, we first downloaded all known sequences of miRNA that have important gene expression regulating functions in *P. vulgaris* from the Plant miRNA Encyclopedia v21.0 database [74]. We then employed the online psRNA Target program to predict the miRNAs that only align and target the PvMYB transcripts using default parameters [75].

### 4.7. Evolutionary Investigation of PvMYB Using the Phylogenetic Tree, Gene Duplication, and Synteny Analysis

The protein sequences of PvMYB were first aligned with Muscle using MEGA X software on default settings [76]. Aligned sequences exported from MEGA X in Fasta format were then used in IQ-tree online software to predict the phylogenetic tree under maximum likelihood using the ultrafast bootstrap approximation v2 (UFBoot2) with 1000 replicates and VT+F+G4 model [77,78,79]. The resulting phylogenetic tree was exported in Newick format and visualized using online ITOL v3 [80]. This alignment was also used to predict repressor PvMYB proteins by assessing the EAR or TLLLRF motifs.

We investigated duplication events in the *R2R3-type PvMYB* genes using the workflow explained in our previous work [66]. Briefly, using Plant Duplication Gene Database (PlantDGD), we identified the similarity and type of duplication for each *PvMYB* gene [38]. Genes with duplicated sequences among *PvMYB* genes were further analyzed individually using BLAST search against the Phytozome v13 database with an e-value cut-off set at <1 × 10^−5^. Then, we selected the top five matches, and within them, the two duplicates with the highest identity were selected as significant matches [29]. Then, collinear blocks were created using the MCScan tool in TBtools software.

To estimate the evolution of *PvMYB* genes based on the ratio of non-synonymous (Ka) to synonymous (Ks) mutation rate (Ka/Ks), we first blasted every PvMYB protein sequence individually on the Phytozome v13 website. Results showing gene pairs with the highest similarity threshold were grouped in a table in a tab-delimited file. The synonymous to non-synonymous substitution rates (Ka/Ks) were calculated via TBtools software. To estimate the divergence time for duplication a million years ago (Mya) for each *PvMYB* gene, we used the following equation: T = Ks/2x (x = 6.56 × 10^−9^).

Multiple collinear scanning toolkits (MCScanX) with the default parameters were used to analyze gene duplication events using synteny. To study the synteny relationships between common bean orthologous *MYB* genes with that of Arabidopsis and rice species, we constructed the homolinear analysis maps using Dual Synteny Plotter software present in TBtools. The genomic information of both *Arabidopsis thaliana* and *Oryza sativa* were downloaded from the Phytozome v13 database. Then, we employed the online SynVisio web tool to visualize the data [31].

## 5. Conclusions

In this study, a comprehensive genome-wide analysis of the common bean R2R3-MYB family genes was performed, and their expression levels were analyzed to elucidate the role of these genes in anthocyanin biosynthesis. In this context, we identified the gene structures, chromosomal locations, gene duplication of R2R3-MYBs, miRNAs related to R2R3 MYBs, and the interaction of these genes with other flavonoid genes for the first time. A total of 160 *PvMYB* genes encoding the R2R3-MYB proteins in the common bean genome were identified using in silico analysis. It was observed that there are at least three and at most eight motifs in the members of the MYB protein family. *PvMYB* genes were found evenly dispersed throughout all chromosomes. We found 44 gene duplication events distributed on 11 chromosomes, suggesting that chromosomal segment duplications can be the key factor for the expansion of the *PvMYB* genes in the common bean genome. Based on the PPI network created using genes involved in the flavonoid/phenylpropanoid pathways and all PvMYBs, probable genes during the seed color development in common beans were selected for qRT-PCR analysis. We observed that *PvMYB92, PvMYB93, PvMYB141, PvMYB142*, and *PvMYB149*, which are the orthologs of tomato skin color-related genes, showed a significant increase in expression in the early pod period. Likewise, we reported that *PvTT8 (bHLH), PvTT2 (PvMYB42), PvMYB113, PvTTG1*, and *PvWD68* have possible roles in the regulation of seed color formation. The result of this study revealed the key importance of *PvMYB* genes in the formation of seed color due to anthocyanin accumulation. In the future, this study will be a valuable information source for the development of bean varieties with different colored seeds containing higher anthocyanin.

## Figures and Tables

**Figure 1 plants-11-03386-f001:**
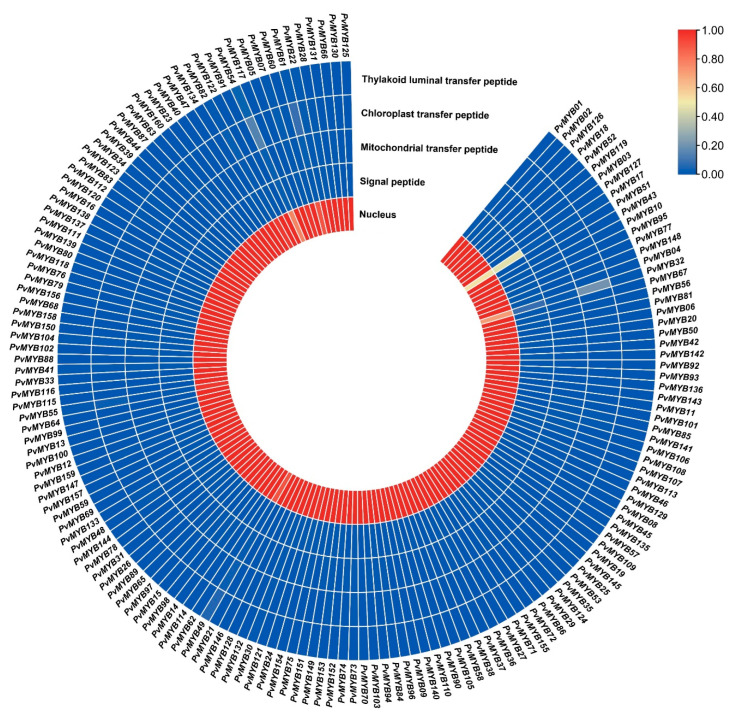
Heatmap illustration for the subcellular localization of PvMYB proteins as predicted by the online TargetP-2.0 server. Data were visualized using TBtools software.

**Figure 2 plants-11-03386-f002:**
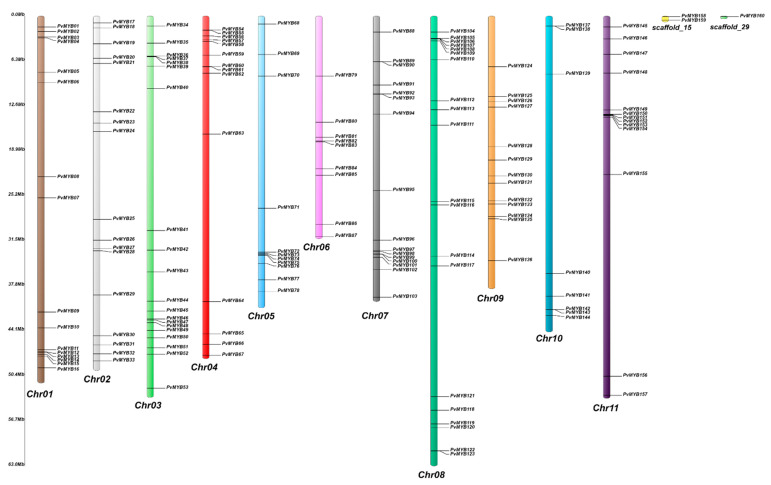
Localization and distribution of 160 *PvMYBs* genes on the chromosome of *P. vulgaris*. The chromosome number is denoted on the bottom of each chromosome, and gene names are displayed on the right side. On the left side, the scale is in megabases (Mb).

**Figure 3 plants-11-03386-f003:**
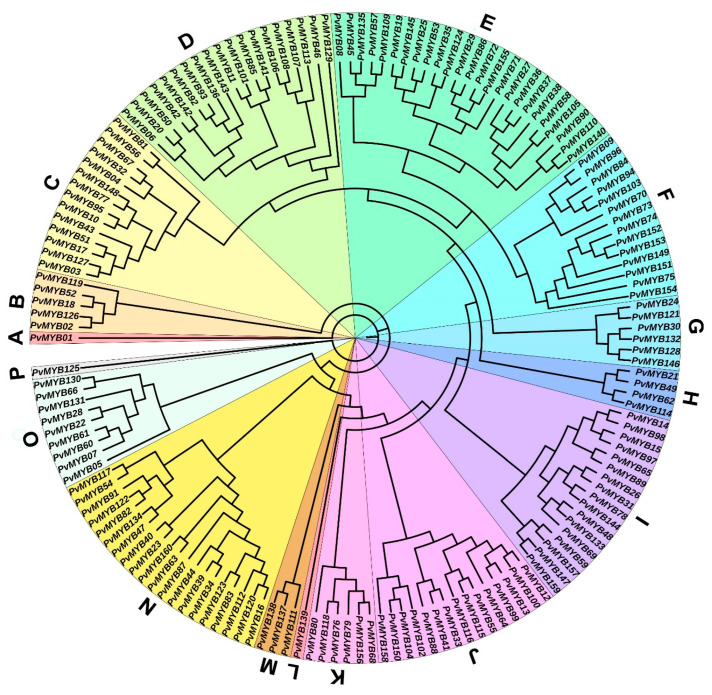
Phylogenetic relationship among PvMYB TFs in *P. vulgaris*. The evolutionary analysis was constructed using IQ-tree online software with 1000 replicas of ultrafast bootstrap settings and VT+F+G4 model. The tree was visualized using online ITOL v3. Lettering and colors on the tree indicate the subgroups (A–P).

**Figure 4 plants-11-03386-f004:**
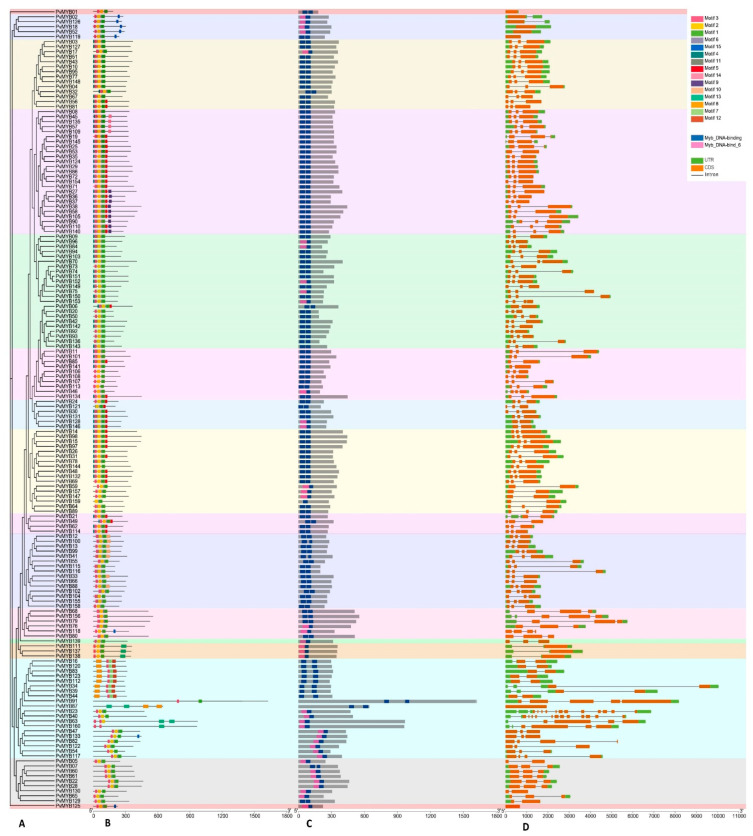
Phylogenetic clustering, motif structure, conserved domains, and exon-intron architecture of 160 *PvMYB* genes. (**A**). Evolutionary tree constructed using IQ-tree online software and clustered into subgroups indicated with different color shades. (**B**). Structure of 15 conserved motifs analyzed through MEME suite tool online. Legends indicate the color of each motif. (**C**). Conserved domain structure of PvMYB TFs in *P. vulgaris*. Two conserved domains were found and presented in blue and pink colors. (**D**). Exon-intron architecture of *PvMYB* genes. The orange boxes indicate exons, the solid black line indicates introns, and the green boxes indicate the non-coding untranslated regions. The below scale bar indicates 100 amino acids.

**Figure 5 plants-11-03386-f005:**
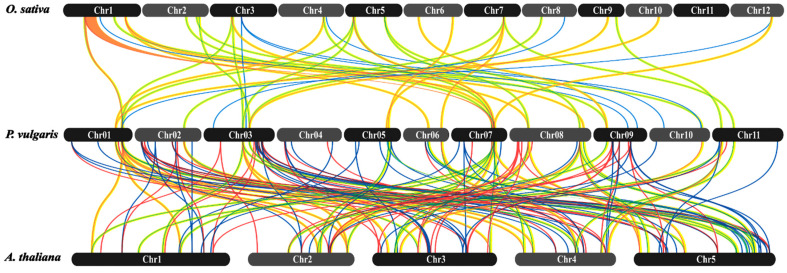
Synteny analysis of *PvMYB* genes between *P. Vulgaris, A. thaliana*, and *O. sativa* representing model plants from dicots and monocots, respectively. Data were analyzed using TBtools software and visualized using the online SynVisio tool. The different color lines represent the synteny relationship of *PvMYB* genes.

**Figure 6 plants-11-03386-f006:**
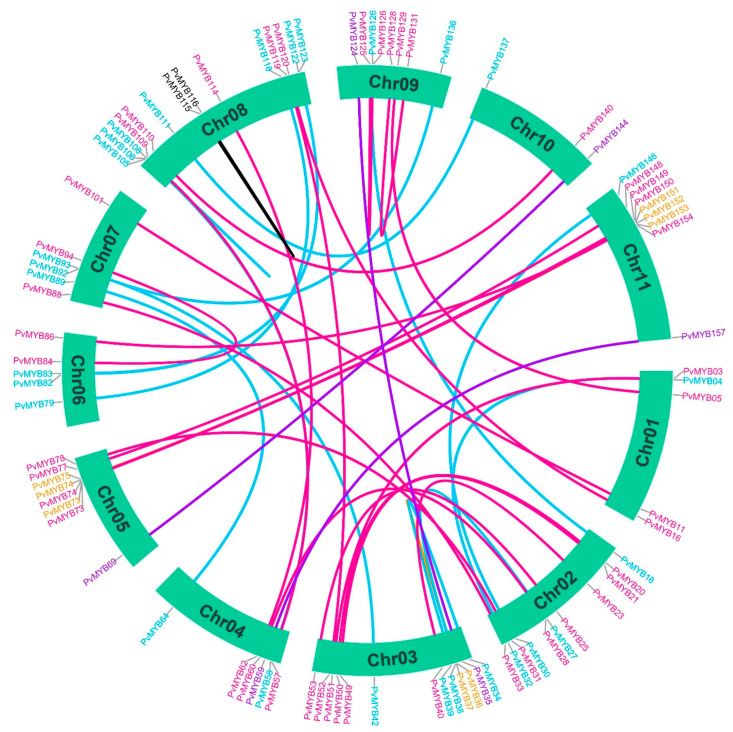
Identified duplicated pairs of R2R3-MYB TFs across all 11 *P. vulgaris* chromosomes. The colored lines denote the duplication type between pairs. Redline indicates whole-genome duplication, yellow indicates tandem duplication, Blackline indicates proximal duplication, purple lines indicate transposed duplication, and cyan-colored lines represent dispersal duplication. Plant Duplication Gene Database (PlantDGD) was used to identify the duplicated gene pairs and the type of duplication. We used TBtools software for the visualization of the data.

**Figure 7 plants-11-03386-f007:**
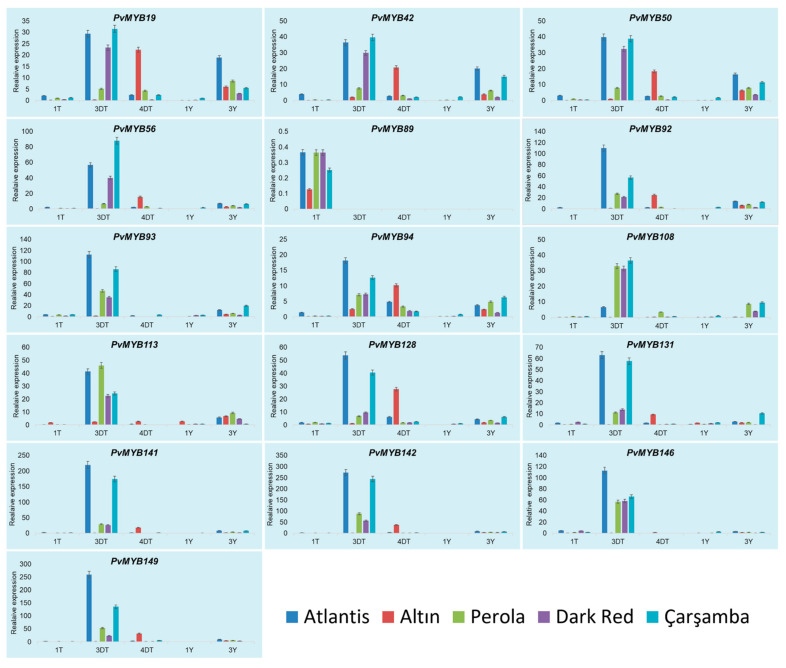
Expression analysis of selected genes 16 *PvMYBs* genes on five different cultivars (Atlantis, Altın, Perola, Dark Red Kidney, and Çarşamba) and on five tissue samples (1DT: First stage of seed development (bud stage), 3DT: Third period of seed development (early pod stage), 4DT: Fourth period of seed development (Pod filling stage), 1Y: leaf sample at seed bud stage, 3Y: leaf sample at seed pod filing stage. Background color shades differentiate *PvMYB* genes from other target genes. Genotype Seminis was used as a control because of white skin color.

**Figure 8 plants-11-03386-f008:**
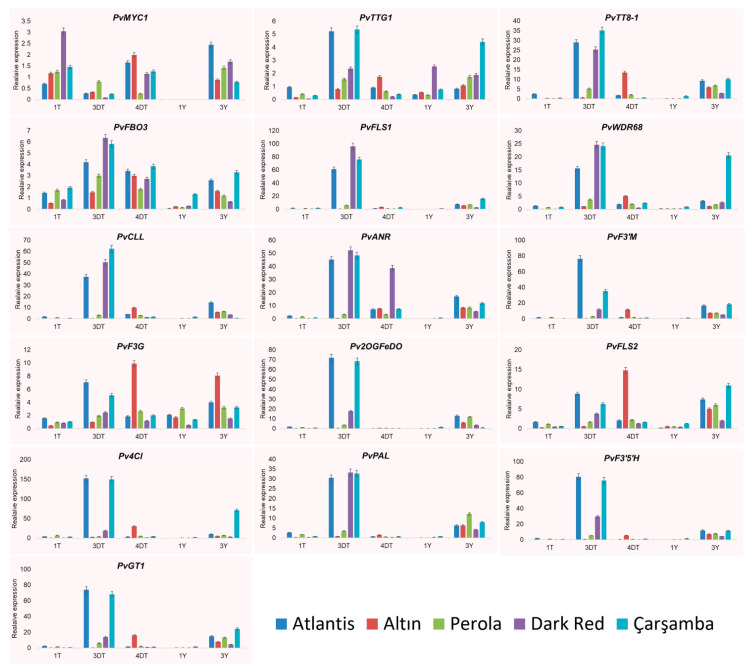
Expression analysis of selected flavonoid biosynthesis related genes having interaction with certain *R2R3-MYB* genes on five different cultivars (Atlantis, Altın, Perola, Dark Red Kidney, and Çarşamba) and on five tissue samples (1DT: First stage of seed development (bud stage), 3DT: Third period of seed development (early pod stage), 4DT: Fourth period of seed development (Pod filling stage), 1Y: leaf sample at seed bud stage, 3Y: leaf sample at seed pod filing stage. Background color shades differentiate *PvMYB* genes from other target genes. Genotype Seminis was used as a control because of white skin color.

**Figure 9 plants-11-03386-f009:**
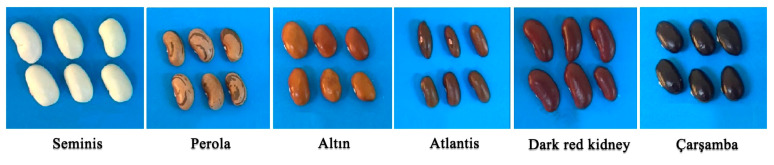
Seeds of different common bean varieties were used in this study. These varieties were selected for their seed coat pigmentations. Seeds used here are commercial varieties used both locally and internationally. From left to right Seminis, Perola, Altın, Atlantis, Dark Red Kidney, and Çarşamba. Seminis was used as a control group for having the lightest seed coat color, and Çarşamba had the darkest seed coat color.

**Table 1 plants-11-03386-t001:** List of identified R2R3-MYB Tfs in *P. vulgaris* and their detailed information.

Gene Name	Phytozome Identifier	Chromosome	Length (bp)	CDS (pb)	Protein Length (A.A)	NCBI Accession Number
*PvMYB01*	Phvul.001G019200.1.v2.1	chr01	610	555	185	XP_007160819.1
*PvMYB02*	Phvul.001G025200.1.v2.1	chr01	1727	831	277	XP_007160886.1
*PvMYB03*	Phvul.001G032100.1.v2.1	chr01	2118	1104	368	XP_007160964.1
*PvMYB04*	Phvul.001G032600.1.v2.1	chr01	2793	903	301	XP_007160971.1
*PvMYB05*	Phvul.001G064600.1.v2.1	chr01	1847	741	247	XP_007161388.1
*PvMYB06*	Phvul.001G071000.1.v2.1	chr01	1613	1092	364	XP_007161463.1
*PvMYB07*	Phvul.001G106800.1.v2.1	chr01	2610	1083	361	XP_007161897.1
*PvMYB08*	Phvul.001G107600.1.v2.1	chr01	1861	1002	334	XP_007161907.1
*PvMYB09*	Phvul.001G161000.1.v2.1	chr01	1965	879	293	XP_007162544.1
*PvMYB10*	Phvul.001G179400.1.v2.1	chr01	2079	996	332	XP_007162775.1
*PvMYB11*	Phvul.001G211700.1.v2.1	chr01	4406	900	300	XP_007163163.1
*PvMYB12*	Phvul.001G215100.1.v2.1	chr01	1290	762	254	XP_007163206.1
*PvMYB13*	Phvul.001G215200.1.v2.1	chr01	1416	804	268	XP_007163207.1
*PvMYB14*	Phvul.001G219000.1.v2.1	chr01	1964	1215	405	XP_007163255.1
*PvMYB15*	Phvul.001G221500.1.v2.1	chr01	2611	1326	442	XP_007163283.1
*PvMYB16*	Phvul.001G240300.1.v2.1	chr01	2438	900	300	XP_007163513.1
*PvMYB17*	Phvul.002G008800.1.v2.1	chr02	1717	1038	346	XP_007156687.1
*PvMYB18*	Phvul.002G015100.1.v2.1	chr02	2145	936	312	XP_007156759.1
*PvMYB19*	Phvul.002G040900.1.v2.1	chr02	2335	975	325	XP_007157076.1
*PvMYB20*	Phvul.002G056900.1.v2.1	chr02	796	555	185	XP_007157272.1
*PvMYB21*	Phvul.002G060500.1.v2.1	chr02	2293	804	268	XP_007157323.1
*PvMYB22*	Phvul.002G083800.1.v2.1	chr02	2419	1389	463	XP_007157609.1
*PvMYB23*	Phvul.002G088900.1.v2.1	chr02	6929	1422	474	XP_007157671.1
*PvMYB24*	Phvul.002G092100.1.v2.1	chr02	1612	696	232	XP_007157714.1
*PvMYB25*	Phvul.002G139500.1.v2.1	chr02	1942	1044	348	XP_007158283.1
*PvMYB26*	Phvul.002G159700.1.v2.1	chr02	2386	951	317	XP_007158529.1
*PvMYB27*	Phvul.002G170000.1.v2.1	chr02	1828	1200	400	XP_007158642.1
*PvMYB28*	Phvul.002G170500.1.v2.1	chr02	2190	1344	448	XP_007158647.1
*PvMYB29*	Phvul.002G221000.1.v2.1	chr02	1525	1089	363	XP_007159239.1
*PvMYB30*	Phvul.002G279000.1.v2.1	chr02	1460	897	299	XP_007159923.1
*PvMYB31*	Phvul.002G292600.1.v2.1	chr02	2737	942	314	XP_007160103.1
*PvMYB32*	Phvul.002G306000.1.v2.1	chr02	1652	912	304	XP_007160258.1
*PvMYB33*	Phvul.002G317000.1.v2.1	chr02	1623	960	320	XP_007160381.1
*PvMYB34*	Phvul.003G013600.1.v2.1	chr03	10049	891	297	XP_007153178.1
*PvMYB35*	Phvul.003G036400.1.v2.1	chr03	1452	939	313	XP_007153451.1
*PvMYB36*	Phvul.003G046200.1.v2.1	chr03	1235	888	296	XP_007153564.1
*PvMYB37*	Phvul.003G046300.1.v2.1	chr03	1124	882	294	XP_007153565.1
*PvMYB38*	Phvul.003G046400.1.v2.1	chr03	3148	1335	445	XP_007153567.1
*PvMYB39*	Phvul.003G054100.1.v2.1	chr03	7172	885	295	XP_007153659.1
*PvMYB40*	Phvul.003G067800.1.v2.1	chr03	5691	1491	497	XP_007153822.1
*PvMYB41*	Phvul.003G118450.1.v2.1	Chr03	2246	939	313	XP_007154150.1
*PvMYB42*	Phvul.003G132100.1.v2.1	chr03	1755	936	312	XP_007154596.1
*PvMYB43*	Phvul.003G148200.1.v2.1	chr03	2020	1086	362	XP_007154791.1
*PvMYB44*	Phvul.003G176800.1.v2.1	chr03	1677	930	310	XP_007155139.1
*PvMYB45*	Phvul.003G190400.1.v2.1	chr03	1520	933	311	XP_007155313.1
*PvMYB46*	Phvul.003G200100.1.v2.1	chr03	1107	588	196	XP_007155422.1
*PvMYB47*	Phvul.003G201300.1.v2.1	chr03	1623	1302	434	XP_007155438.1
*PvMYB48*	Phvul.003G203900.1.v2.1	chr03	1668	1110	370	XP_007155468.1
*PvMYB49*	Phvul.003G214300.1.v2.1	chr03	1771	960	320	XP_007155582.1
*PvMYB50*	Phvul.003G222400.1.v2.1	chr03	1548	567	189	XP_007155685.1
*PvMYB51*	Phvul.003G232300.1.v2.1	chr03	1545	972	324	XP_007155794.1
*PvMYB52*	Phvul.003G240200.1.v2.1	chr03	1671	870	290	XP_007155887.1
*PvMYB53*	Phvul.003G284000.1.v2.1	chr03	1582	1041	347	XP_007156413.1
*PvMYB54*	Phvul.004G011400.1.v2.1	chr04	2184	879	293	XP_007151013.1
*PvMYB55*	Phvul.004G012000.1.v2.1	chr04	3691	660	220	XP_007151019.1
*PvMYB56*	Phvul.004G024200.1.v2.1	chr04	1690	996	332	XP_007151175.1
*PvMYB57*	Phvul.004G028500.1.v2.1	chr04	1890	954	318	XP_007151227.1
*PvMYB58*	Phvul.004G029800.1.v2.1	chr04	3046	1227	409	XP_007151240.1
*PvMYB59*	Phvul.004G046000.1.v2.1	chr04	3445	1050	350	XP_007151435.1
*PvMYB60*	Phvul.004G053500.1.v2.1	chr04	2056	1131	377	XP_007151515.1
*PvMYB61*	Phvul.004G053600.1.v2.1	chr04	2104	1158	386	XP_007151516.1
*PvMYB62*	Phvul.004G057800.1.v2.1	chr04	1357	828	276	XP_007151568.1
*PvMYB63*	Phvul.004G090900.1.v2.1	chr04	6608	2910	970	XP_007151972.1
*PvMYB64*	Phvul.004G116500.1.v2.1	chr04	2635	870	290	XP_007158818.1
*PvMYB65*	Phvul.004G144900.1.v2.1	chr04	3058	693	231	XP_007152283.1
*PvMYB66*	Phvul.004G156154.1.v2.1	Chr04	1483	903	301	XP_007152615.1
*PvMYB67*	Phvul.004G173500.1.v2.1	chr04	1282	810	270	XP_007152944.1
*PvMYB68*	Phvul.005G012900.1.v2.1	chr05	4278	1536	512	XP_007148771.1
*PvMYB69*	Phvul.005G047400.1.v2.1	chr05	1644	969	323	XP_007149171.1
*PvMYB70*	Phvul.005G060000.1.v2.1	chr05	2932	1212	404	XP_007149313.1
*PvMYB71*	Phvul.005G087400.1.v2.1	chr05	1861	1125	375	XP_007149649.1
*PvMYB72*	Phvul.005G109100.1.v2.1	chr05	1316	972	324	XP_007149908.1
*PvMYB73*	Phvul.005G109700.1.v2.1	chr05	1454	981	327	XP_007149915.1
*PvMYB74*	Phvul.005G109800.1.v2.1	chr05	3191	684	228	XP_007149916.1
*PvMYB75*	Phvul.005G109900.1.v2.1	chr05	4174	699	233	XP_007149917.1
*PvMYB76*	Phvul.005G115500.1.v2.1	chr05	3778	1437	479	XP_007149977.1
*PvMYB77*	Phvul.005G131300.1.v2.1	chr05	1935	1014	338	XP_007150152.1
*PvMYB78*	Phvul.005G157600.1.v2.1	chr05	2073	786	262	XP_007150497.1
*PvMYB79*	Phvul.006G020200.1.v2.1	chr06	5751	1578	526	XP_007146191.1
*PvMYB80*	Phvul.006G045300.1.v2.1	chr06	2290	1539	513	XP_007146493.1
*PvMYB81*	Phvul.006G061600.1.v2.1	chr06	1169	978	326	XP_007146695.1
*PvMYB82*	Phvul.006G064600.1.v2.1	chr06	5300	1311	437	XP_007146726.1
*PvMYB83*	Phvul.006G065700.1.v2.1	chr06	2767	945	315	XP_007146740.1
*PvMYB84*	Phvul.006G105200.1.v2.1	chr06	1229	648	216	XP_007147213.1
*PvMYB85*	Phvul.006G114800.1.v2.1	chr06	1622	846	282	XP_007147330.1
*PvMYB86*	Phvul.006G192900.1.v2.1	chr06	1577	1092	364	XP_007148253.1
*PvMYB87*	Phvul.006G217200.1.v2.1	chr06	1941	1941	647	XP_007148540.1
*PvMYB88*	Phvul.007G028700.1.v2.1	chr07	1672	918	306	XP_007142925.1
*PvMYB89*	Phvul.007G069200.1.v2.1	chr07	2444	813	271	XP_007143397.1
*PvMYB90*	Phvul.007G069400.1.v2.1	chr07	3042	966	322	XP_007143400.1
*PvMYB91*	Phvul.007G093100.1.v2.1	chr07	8179	4878	1626	XP_007143686.1
*PvMYB92*	Phvul.007G099900.1.v2.1	chr07	1116	837	279	XP_007143768.1
*PvMYB93*	Phvul.007G100100.1.v2.1	chr07	1333	765	255	XP_007143770.1
*PvMYB94*	Phvul.007G108500.1.v2.1	chr07	2434	804	268	XP_007143866.1
*PvMYB95*	Phvul.007G147600.1.v2.1	chr07	2077	939	313	XP_007144336.1
*PvMYB96*	Phvul.007G192900.1.v2.1	chr07	1061	804	268	XP_007144899.1
*PvMYB97*	Phvul.007G206200.1.v2.1	chr07	2040	1206	402	XP_007145057.1
*PvMYB98*	Phvul.007G208400.1.v2.1	chr07	2113	1338	446	XP_007145085.1
*PvMYB99*	Phvul.007G211800.1.v2.1	chr07	1765	777	259	XP_007145121.1
*PvMYB100*	Phvul.007G211900.1.v2.1	chr07	1202	846	282	XP_007145122.1
*PvMYB101*	Phvul.007G215800.1.v2.1	chr07	4034	1035	345	XP_007145164.1
*PvMYB102*	Phvul.007G231800.1.v2.1	chr07	1407	864	288	XP_007145351.1
*PvMYB103*	Phvul.007G273400.1.v2.1	chr07	2248	765	255	XP_007145850.1
*PvMYB104*	Phvul.008G028000.1.v2.1	chr08	1673	786	262	XP_007139421.1
*PvMYB105*	Phvul.008G038000.1.v2.1	chr08	3435	1146	382	XP_007139534.1
*PvMYB106*	Phvul.008G038200.1.v2.1	chr08	1070	693	231	XP_007139536.1
*PvMYB107*	Phvul.008G038400.1.v2.1	chr08	2275	642	214	XP_007139538.1
*PvMYB108*	Phvul.008G038600.1.v2.1	chr08	1081	753	251	XP_007139540.1
*PvMYB109*	Phvul.008G041500.1.v2.1	chr08	1506	975	325	XP_007139579.1
*PvMYB110*	Phvul.008G067300.1.v2.1	chr08	2643	927	309	XP_007139892.1
*PvMYB111*	Phvul.008G102300.1.v2.1	chr08	3132	1071	357	XP_007140319.1
*PvMYB112*	Phvul.008G107000.1.v2.1	chr08	2224	861	287	XP_007140381.1
*PvMYB113*	Phvul.008G113300.1.v2.1	chr08	1965	672	224	XP_007140451.1
*PvMYB114*	Phvul.008G146900.1.v2.1	chr08	1056	807	269	XP_007140849.1
*PvMYB115*	Phvul.008G155700.1.v2.1	chr08	3582	603	201	XP_007140962.1
*PvMYB116*	Phvul.008G155900.1.v2.1	chr08	4724	600	200	XP_007140964.1
*PvMYB117*	Phvul.008G156100.1.v2.1	chr08	4583	1191	397	XP_007140966.1
*PvMYB118*	Phvul.008G205000.1.v2.1	chr08	1443	990	330	XP_007141541.1
*PvMYB119*	Phvul.008G222600.1.v2.1	chr08	720	720	240	XP_007141751.1
*PvMYB120*	Phvul.008G226600.1.v2.1	chr08	2172	918	306	XP_007141798.1
*PvMYB121*	Phvul.008G233800.1.v2.1	chr08	1071	612	204	XP_007141884.1
*PvMYB122*	Phvul.008G262700.1.v2.1	chr08	3968	1110	370	XP_007142225.1
*PvMYB123*	Phvul.008G262900.1.v2.1	chr08	2006	915	305	XP_007142227.1
*PvMYB124*	Phvul.009G031200.1.v2.1	chr09	1512	1008	336	XP_007136252.1
*PvMYB125*	Phvul.009G062700.1.v2.1	chr09	675	675	225	XP_007136655.1
*PvMYB126*	Phvul.009G068000.1.v2.1	chr09	2081	789	263	XP_007136715.1
*PvMYB127*	Phvul.009G075000.1.v2.1	chr09	1808	1035	345	XP_007136795.
*PvMYB128*	Phvul.009G119900.1.v2.1	chr09	1315	780	260	XP_007137351.1
*PvMYB129*	Phvul.009G133700.1.v2.1	chr09	1643	996	332	XP_007134825.1
*PvMYB130*	Phvul.009G151000.1.v2.1	chr09	1061	924	308	XP_007137516.1
*PvMYB131*	Phvul.009G158200.1.v2.1	chr09	1660	957	319	XP_007137729.1
*PvMYB132*	Phvul.009G174900.1.v2.1	chr09	1707	1068	356	XP_007137818.1
*PvMYB133*	Phvul.009G177100.1.v2.1	chr09	1640	1338	446	XP_007138025.1
*PvMYB134*	Phvul.009G185950.1.v2.1	Chr09	2431	1347	449	XP_007138059.1
*PvMYB135*	Phvul.009G187700.1.v2.1	chr09	1713	948	316	XP_007138187.1
*PvMYB136*	Phvul.009G228200.1.v2.1	chr09	2842	576	192	XP_007138671.1
*PvMYB137*	Phvul.010G009800.1.v2.1	chr10	3638	1059	353	XP_007133991.1
*PvMYB138*	Phvul.010G009900.1.v2.1	chr10	3098	1047	349	XP_007133992.1
*PvMYB139*	Phvul.010G053200.1.v2.1	chr10	2065	948	316	XP_007134506.1
*PvMYB140*	Phvul.010G096400.1.v2.1	chr10	2768	837	279	XP_007135039.1
*PvMYB141*	Phvul.010G115500.1.v2.1	chr10	1194	879	293	XP_007135274.1
*PvMYB142*	Phvul.010G130500.1.v2.1	chr10	1279	879	293	XP_007135450.1
*PvMYB143*	Phvul.010G130600.1.v2.1	chr10	1506	786	262	XP_007135451.1
*PvMYB144*	Phvul.010G137500.1.v2.1	chr10	1804	1041	347	XP_007135535.1
*PvMYB145*	Phvul.011G019200.1.v2.1	chr11	1516	969	323	XP_007131513.1
*PvMYB146*	Phvul.011G034900.1.v2.1	chr11	1416	762	254	XP_007131703.1
*PvMYB147*	Phvul.011G059800.1.v2.1	chr11	2344	984	328	XP_007132019.1
*PvMYB148*	Phvul.011G084500.1.v2.1	chr11	2090	936	312	XP_007132317.1
*PvMYB149*	Phvul.011G109400.1.v2.1	chr11	1593	774	258	XP_007132609.1
*PvMYB150*	Phvul.011G109500.1.v2.1	chr11	4958	690	230	XP_007144143.1
*PvMYB151*	Phvul.011G109600.1.v2.1	chr11	1458	972	324	XP_007132610.1
*PvMYB152*	Phvul.011G109700.1.v2.1	chr11	1493	975	325	XP_007132611.1
*PvMYB153*	Phvul.011G109800.1.v2.1	chr11	1301	675	225	XP_007132612.1
*PvMYB154*	Phvul.011G110500.1.v2.1	chr11	1300	957	319	XP_007132613.1
*PvMYB155*	Phvul.011G115230.1.v2.1	Chr11	1292	789	263	XP_007132622.1
*PvMYB156*	Phvul.011G191300.1.v2.1	chr11	4855	1665	555	XP_007133583.1
*PvMYB157*	Phvul.011G212000.1.v2.1	chr11	2698	915	305	XP_007133825.1
*PvMYB158*	Phvul.L002743.1.v2.1	scaffold_15	1663	717	239	XP_007150291.1
*PvMYB159*	Phvul.L009043.1.v2.1	scaffold_15	2870	834	278	XP_007150366.1
*PvMYB160*	Phvul.L001860.1.v2.1	scaffold_29	5338	2892	964	XP_007144020.1

## Data Availability

Not applicable.

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
