# Peer review of "Investigation and Expression Analysis of R2R3-MYBs and Anthocyanin Biosynthesis-Related Genes during Seed Color Development of Common Bean (Phaseolus vulgaris)"

_plants, 2022, doi:10.3390/plants11233386_

Round 1

Reviewer 1 Report

1. Please provide the source of the varieties used in this article.

2. please give the details of the planting of beans. For example: the plot size, which type soils used in this study, how much soil does each pot contain?

3. in line 571-573, please give the time of different stage, for example, about (1DT, 70d)

4. The conclusion is not very good, and there is no better summary of the research results of this paper. It is suggested that the author summarize the main results of this paper with (1), (2), (3)

Author Response

Thank you very much for your valuable suggestions. According to your proposal, we organized the whole manuscript, and we made the required corrections. Track-changed version of the manuscript was also added.

Point 1: Please provide the source of the varieties used in this article.

Response 1: Following statement was added to the manuscript.

“The common bean varieties used in this study were obtained from the Middle Black Sea Transitional Zone Agricultural Research Institute”

Point 2: please give the details of the planting of beans. For example: the plot size, which type soils used in this study, how much soil does each pot contain?

Response 2: Following statement was added to the manuscript.

“Seeds were then sown in clean plastic pots (5.2lt) filled with soil-free vermiculite (approximately 5 kg for each pots) to ensure uniformity and the absence of substrate for microbial growth. In order to ensure a homogeneous germination in all pots, 3 seeds were planted in each pot and after homogeneity was achieved in the hypocotyl stage, one plant was left in each pot. Pots were irrigated once a week with a Hoagland's solution. The seeds were grown in a controlled growth chamber set at 24°C with a photoperiod of 16 hours light and 8 hours dark and humidity of 60-65%. “

Point 3. in line 571-573, please give the time of different stage, for example, about (1DT, 70d)

Response 3: Following statement was added to the manuscript.

“Samples were collected at three different stages, including bud (1DT, 40 days after germination) and accompanied leaves (1Y),  early pod (3DT, 55 days after germination) and accompanied leaves (3Y), and pod fill (4DT, 65 days after germination), were collected from all cultivars (Figure S4).”

Point 4. The conclusion is not very good, and there is no better summary of the research results of this paper. It is suggested that the author summarize the main results of this paper with (1), (2), (3)

Response 4: Conclusion part was revised.

Reviewer 2 Report

This manuscript provides the result of essentially mainly bioinformatics analysis of selected group (R2R3-MYB) of transcription factors in common bean. Resulting in gene structures, chromosomal placements, gene duplication and miRNAs associated with R2R3-MYBs.

The experimental part is based on the study of 16 selected R2R3-MYB genes and 16 phenylpropanoid pathway genes using qRT-PCR in 5 different tissues at different developmental stages. However there is no demonstrated relationship between selected TFs and phenylpropanoid pathway genes. Finaly, there is no demonstrated relationship in their expression and seed coat pigmentation, since solely qPCR is not sufficient. Further experiments, including genetics (crosses) are needed to clarify this.

Discussion is unfocussed and contain numerous parts which are not directly linked to the study.

Author Response

Thank you very much for your valuable suggestions. According to your proposal, we organized the whole manuscript, and we made the required corrections.Track-change version of the manuscript was also submitted.

Point 1:  This manuscript provides the result of essentially mainly bioinformatics analysis of selected group (R2R3-MYB) of transcription factors in common bean. Resulting in gene structures, chromosomal placements, gene duplication and miRNAs associated with R2R3-MYBs.

The experimental part is based on the study of 16 selected R2R3-MYB genes and 16 phenylpropanoid pathway genes using qRT-PCR in 5 different tissues at different developmental stages. However there is no demonstrated relationship between selected TFs and phenylpropanoid pathway genes. Finaly, there is no demonstrated relationship in their expression and seed coat pigmentation, since solely qPCR is not sufficient. Further experiments, including genetics (crosses) are needed to clarify this.

Response 1: We selected 32 genes by looking at the Protein-Protein interaction network. As you can appreciate, it is very difficult to evaluate such a large number of genes ( 160 R2R3 MYB and 46 Flavonoid biosythesis related genes) with qRT-PCR, and we preferred to work with genes included in this interaction network.

Point 2:  Discussion is unfocussed and contain numerous parts which are not directly linked to the study.

Response 2: Whole manuscript and especially discussion part were revised. Track change version of the manuscript was also submitted.

Reviewer 3 Report

MYB proteins are key regulators controlling the expression of anthocyanin related genes in plants. Kavas et al. proposed the genome wide identification, evolution, and expression of R2R3-MYBs in P. vulgaris. The study has been supported by computational and molecular analyses which seem to be optimal. In my judgement, this manuscript can be considered for publication in Plants after some necessary revisions as follows.

1. In the Introduction section there is extremely basic information. But this section does not cover all the research activities accomplished in this study. For examples, authors have identified different types of duplications in different genes, but these genes were not mentioned at all in the Introduction section. Moreover, the words describing the importance of something are not proper. For an instance, in “most crucial legume crop” in line 47, I am unable to digest the information whether P. vulgaris could be matter of survival globally. I would suggest extensive revisions to make information flow throughout the manuscript in an appropriate way as such mistakes were also seen in other sections.

2. The aims and objectives of this study are not well explained.

3. In phylogenetic analysis, authors claimed that subgroup P is the largest group containing 26 PvMYBs. But, I could not find P as the largest one. In fact, there is only one gene in P subgroup, and the subgroup E might be the largest one having 25 PvMYBs. Authors need to check and rearrange the results.

4. Figures 4 and 7 are not clear enough to distinguish the variations, and authors should make them large with high resolution or separate them into more figures.

5. Citations for references are missing in lines 239-242.

6. Figures 8 and 9 can be moved to supplementary materials.

7. In some places the scientific names were not printed in italic.

Author Response

Response to Reviewer 3 Comments

Thank you very much for your valuable suggestions. According to your proposal, we organized the whole manuscript, and we made the required corrections. Track-change version of the manuscript was also submitted.

Point 1:  In the Introduction section there is extremely basic information. But this section does not cover all the research activities accomplished in this study. For examples, authors have identified different types of duplications in different genes, but these genes were not mentioned at all in the Introduction section. Moreover, the words describing the importance of something are not proper. For an instance, in “most crucial legume crop” in line 47, I am unable to digest the information whether P. vulgaris could be matter of survival globally. I would suggest extensive revisions to make information flow throughout the manuscript in an appropriate way as such mistakes were also seen in other sections.

Response 1: Whole manuscript and especially discussion part were revised. Track change version of the manuscript was also submitted.

Point 2. The aims and objectives of this study are not well explained.

Response 2: This part was added to the manuscript.

“In this study, a comprehensive genome-wide analysis of the common bean R2R3-MYB family genes was performed, and their expression levels were analyzed to elucidate the role of these genes in anthocyanin biosynthesis. In this context, we identified the gene structures, chromosomal locations, gene duplication of R2R3-MYBs, miRNAs related to R2R3 MYBs, and the interaction of these genes with other flavonoid genes for the first time. A total of 160 PvMYB genes encoding the R2R3-MYB proteins in the common bean genome were identified by using in silico analysis”

Point 3.  In phylogenetic analysis, authors claimed that subgroup P is the largest group containing 26 PvMYBs. But, I could not find P as the largest one. In fact, there is only one gene in P subgroup, and the subgroup E might be the largest one having 25 PvMYBs. Authors need to check and rearrange the results.

Response 3: We would like to thank you for your very important warning. We interpreted the results according to the old version figure, but necessary corrections were made.

Point 4. Figures 4 and 7 are not clear enough to distinguish the variations, and authors should make them large with high resolution or separate them into more figures.

Response: In fact, we are aware that the figures are very large, but it is very difficult to create a meaningful figure, especially when the figure 4 in the phylogenetic tree is divided. However, in line with your suggestion, Figure 7 and Figure 8 were created by dividing Figure 7 and added to the article.

Point 5. Citations for references are missing in lines 239-242.

Response 5: Citation was added.

Point 6. Figures 8 and 9 can be moved to supplementary materials.

Response 6: Figure 9 was moved to Supplementary files.

Point 7. In some places the scientific names were not printed in italic.

Response 7: Required corrections were made.

Round 2

Reviewer 2 Report

As mentioned in my first review, this is solely bioinformatic analysis of MYB transcription factory family in common bean. There is no demonstrated relationship between selected TFs and phenylpropanoid pathway genes. This can not be improved by just revision but additional experiments.

Author Response

Dear reviewer, first of all, thank you very much for your suggestion. 

The main purpose of this study is to identify the R2R3 type MYB genes found in the common  bean genome using bioinformatics tools. In addition, we also performed qPCR analyzes to understand the roles of these genes in seed color formation with qPCR. Since there are 160 R2R3-type MYBs in total and not all of them are easy to analyze by qPCR, we selected some of them for these analyses. In order to have a scientific basis for our selection, we prepared a protein-protein interaction network and analyzed some MYBs and flavonoid pathway genes in this network by qPCR. Therefore, our aim here is not to show an existing relationship between selected flavonoid genes and MYBs, but to determine the effect of 32 genes on bean seed color formation. As a result of these analyzes, we determined that especially PvTT8 (bHLH), PvTT2 (PvMYB42), PvMYB113, PvTTG1, and PvWD68 genes may be effective in this mechanism. If our aim was to show the interaction between these proteins, we would do a Co-Immunoprecipitation analysis, but the probability of this study being performed with 32 proteins is very low. In genome-wide analyses, some genes are generally selected and their activity evaluated by qPCR. Numerous articles published in MDPI journals include studies conducted in this way. In fact, many articles, such as the articles I have given below, have only in silico analyzes and even qPCR analyzes have not been performed.

Zhu, L.; Ding, Y.; Wang, S.; Wang, Z.; Dai, L. Genome-Wide Identification, Characterization, and Expression Analysis of CHS Gene Family Members in Chrysanthemum nankingenseGenes 202213, 2145. https://doi.org/10.3390/genes13112145

Islam, M.A.U.; Nupur, J.A.; Khalid, M.H.B.; Din, A.M.U.; Shafiq, M.; Alshegaihi, R.M.; Ali, Q.; Ali, Q.; Kamran, Z.; Manzoor, M.; Haider, M.S.; Shahid, M.A.; Manghwar, H. Genome-Wide Identification and In Silico Analysis of ZF-HD Transcription Factor Genes in Zea mays L. Genes 202213, 2112. https://doi.org/10.3390/genes13112112